# Bempedoic Acid Restores Liver H_2_S Production in a Female Sprague-Dawley Rat Dietary Model of Non-Alcoholic Fatty Liver

**DOI:** 10.3390/ijms24010473

**Published:** 2022-12-28

**Authors:** Núria Roglans, Elena Fauste, Roger Bentanachs, Ana M. Velázquez, Madelin Pérez-Armas, Cristina Donis, María I. Panadero, Marta Alegret, Paola Otero, Carlos Bocos, Juan C. Laguna

**Affiliations:** 1Department of Pharmacology, Toxicology and Therapeutic Chemistry, School of Pharmacy and Food Science, University of Barcelona, Av. Joan XXIII 27–31, 08028 Barcelona, Spain; 2Spanish Biomedical Research Centre in Physiopathology of Obesity and Nutrition (CIBEROBN), Instituto de Salud Carlos III (ISCIII), 28029 Madrid, Spain; 3Institute of Biomedicine, University of Barcelona, 08028 Barcelona, Spain; 4Facultad de Farmacia, Universidad San Pablo-CEU, CEU Universities, Montepríncipe, Boadilla del Monte, 28668 Madrid, Spain

**Keywords:** NAFLD, FXR, mTORC1, S6K1, PGC1α, fructose, high-fat diet

## Abstract

We previously demonstrated that treatment with BemA (bempedoic acid), an inhibitor of ATP citrate lyase, significantly reduces fatty liver in a model of liver steatosis (HFHFr—female Sprague-Dawley rat fed a high-fat high-fructose diet). Since the hepatic production of the gasotransmitter H_2_S is impaired in liver disorders, we were interested in determining if the production of H_2_S was altered in our HFHFr model and whether the administration of BemA reversed these changes. We used stored liver samples from a previous study to determine the total and enzymatic H_2_S production, as well as the expression of CBS (cystathionine β-synthase), CSE (cystathionine γ-lyase), and 3MST (3-mercaptopiruvate sulfurtransferase), and the expression/activity of FXR (farnesoid X receptor), a transcription factor involved in regulating CSE expression. Our data show that the HFHFr diet reduces the total and enzymatic production of liver H_2_S, mainly by decreasing the expression of CBS and CSE. Furthermore, BemA treatment restored H_2_S production, increasing the expression of CBS and CSE, providing evidence for the involvement of FXR transcriptional activity and the mTORC1 (mammalian target of rapamycin1)/S6K1 (ribosomal protein S6 kinase beta-1)/PGC1α (peroxisome proliferator receptor gamma coactivator1α) pathway.

## 1. Introduction

H_2_S (hydrogen sulfide) is the latest addition to the group of so-called gasotransmitters, which include NO (nitric oxide) and CO (carbon monoxide) [1]. It is involved in several diverse physiological and pathophysiological processes, including obesity, hypertension, type 2 Diabetes Mellitus, and NAFLD (non-alcoholic fatty liver disease) [2,3,4]. H_2_S is able to modulate the activity of many proteins via post-translational modification (persulfidation) and is also a potent reducing agent, contributing to cellular redox homeostasis [1]. H_2_S is produced within the transsulfuration pathway by the activity of three enzymes: CBS (cystathionine β-synthase), CSE (cystathionine γ-lyase), and 3MST (3-mercaptopiruvate sulphurtransferase). CBS and CSE are highly expressed in the liver, the main organ responsible for H_2_S production, while 3MST is mainly expressed in glial cells [1,4]. CBS activity is controlled by the availability of substrates and post-translational modifications [4], while CSE activity is controlled at the level of gene transcription by the PERK (protein kinase RNA-like endoplasmic reticulum kinase) branch of the ERSR (endoplasmic reticulum stress response) [5], as well as by several transcription factors (SREBP1c—sterol response binding protein 1c [6]) and receptors (FXR—farnesoid X receptor [7], GPBAR-1—G-protein-coupled bile acid receptor 1 [8]).

NAFLD currently affects 25% of the human population. From the initial stage of simple steatosis (caused by the accumulation of triglycerides in the form of lipid droplets), a quarter of patients will go on to develop NASH (steatohepatitis), which increases the risk of cirrhosis and hepatocellular carcinoma [9,10]. Nowadays, there is no approved drug therapy for NAFLD. Different types of HFDs (high-fat diets) are used in experimental models to induce NAFLD [11]. Furthermore, by increasing the expression and transcriptional activity of SREBP1c, the supplementation of an HFD can reduce the liver expression of CSE in mice [6]. Very recently, we showed that the supplementation of a 10% *w*/*v* fructose solution in Sprague-Dawley rats also resulted in a reduction in the liver expression of CSE and, consequently, in the production of hepatic H_2_S [12,13].

BemA (bempedoic acid) is an inhibitor of ACLY (ATP citrate lyase). It was recently approved for the treatment of primary hypercholesterolemia or mixed dyslipidemia in adults, alone or in combination with other lipid-lowering therapies such as statins or ezetimibe [14]. Through its key position in the biosynthetic route of simple lipids, liver ACLY activity controls not only cholesterol, but also fatty acid synthesis [15], thereby making BemA a good drug candidate for treating liver steatosis, the early stage of NAFLD. Recently, we described a new dietary model of fatty liver without the concomitant induction of liver inflammation, obesity, and clear signs of whole-body insulin resistance. This model was developed by administering a solid HFD with a plant-based origin (without added cholesterol) and a 10% w/v fructose solution as a beverage (HFHFr diet) to female Sprague-Dawley rats [16]. Using this model, we recently demonstrated that BemA administration strongly reverses hepatic triglyceride deposition [17,18]. Thus, based on the abovementioned scientific knowledge about liver CSE expression and H_2_S production, we were interested in determining whether the HFHFr dietary intervention in female rats was able to induce not only fatty liver, but also a reduction in liver H_2_S production and, if so, to test whether BemA administration could reverse these changes.

The data reported in the present work, using stored liver samples from our previous study [17,18], clearly show that the HFHFr diet reduces liver CBS and CSE expression, as well as H_2_S production in female Sprague-Dawley rats. Furthermore, these changes are fully reversed by BemA treatment. We provide possible molecular mechanisms responsible for these changes, involving FXR, PGC-1α (peroxisome proliferator receptor gamma coactivator-1α), and mTORC1 (mammalian target of rapamycin 1).

## 2. Results

### 2.1. The Reduction of H_2_S Production in the Steatotic Livers of Female Sprague-Dawley Rats Is Reversed by Bempedoic Acid Administration, Restoring CBS and CSE Expression

The livers of control female Sprague-Dawley rats showed the ability to produce measurable amounts of H_2_S that was mainly derived from enzymatic activity (88%, Figure 1). After a three-month feeding period with the HFHFr diet, the livers of Sprague-Dawley rats showed a reduction in total H_2_S production (×0.69), mainly due to a decrease in the H_2_S synthetized by enzymatic activity (×0.63) (Figure 1).

BemA administration to the female HFHFr Sprague-Dawley rats reversed these changes, restoring the total and enzymatic production of liver H_2_S (Figure 1). As liver H_2_S is mainly produced by enzymatic activity, we looked at the expression of the three enzymes involved in the generation of H_2_S in liver tissue: CBS, CSE, and 3MST. As can be seen in Figure 2, the HFHFr-fed rats showed a significant decrease in the expression of *cbs* (×0.47 vs. CT) and *cse* (×0.68 vs. CT) genes, but no changes in the expression of the *3mst* gene were observed. Conversely, BemA administration to the HFHFr-fed rats increased the liver expression of both *cbs* and *cse* genes, almost to control values, without affecting the expression of the *3mst* gene. As CSE is the main enzyme responsible for the enzymatic production of liver H_2_S [4], we looked at possible changes in the expression/activity of transcription factors known to control the expression of the *cse* gene associated with the consumption of the HFHFr diet and/or BemA administration.

### 2.2. BemA Administration Increases the Expression and Activity of FXR in the Livers of HFHFr-Fed Female Sprague-Dawley Rats

As FXR is involved in the control of CSE liver expression [7], we first determined the presence of the FXR protein in nuclear extracts of the liver samples as an indirect measure of FXR activity. As shown in Figure 3A, the HFHFr samples showed a significant increase in the relative nuclear presence of FXR (×2.2), which was further increased with BemA administration (×2.9 vs. CT, ×1.34 vs. HFHFr). We next determined the expression of the *cyp7a1* gene, which is under FXR control [19]. Levels of the *cyp7a1* mRNA (Figure 3B) and protein (Figure 3C) were markedly reduced in liver samples of both the HFHFr-fed and BemA-treated rats, pointing to increased activity of FXR in the livers of these animals. Since the decrease in the *cse* mRNA levels in the HFHF liver samples was not in accordance with an increase in FXR transcriptional activity (see Figure 2), we determined the expression of several genes that are under FXR transcriptional control. As shown in Figure 3B and Figure 4, the liver samples from BemA-treated rats showed significant increases vs. control values in the mRNA levels of *shp* (×4.2), *cyp8b1* (×1.9), and *abcg5* (×4.2), while the mRNA levels of *cyp7a1* (×0.1) and *mafg* (×0.6) were decreased, clearly pointing to increased FXR transcriptional activity. SHP (small heterodimerization partner), an orphan nuclear receptor without LBD, is directly involved in the inhibition of *cyp7a1* and *mafg* expression [19]. On the contrary, in the liver samples of HFHFr-fed rats, mRNA levels were either unmodified (*shp*, *cyp8b1*, and *mafg*) or even decreased (*abcg*5- × 0.2-, *cyp7a1*- × 0.1-), showing no concordance with a hypothetical increase in FXR transcriptional activity.

Due to the observed effects on enzymes involved in bile acid synthesis, we determined the concentration of total bile acids in the liver tissue and serum, as well as the serum concentration of FGF15 (fibroblast growth factor 15). FGF15 is produced in the ileum by the bile acid activation of ileal FXR [19]. As shown in Figure 5, although HFHFr-fed animals did not show significant changes in the liver and serum total bile acid concentrations, they presented a significant reduction in the production of ileal FGF15/19 (×0.9 vs. control). BemA treatment restored the serum FGF15/19 concentration and even significantly increased the liver bile acid concentration (×4.2 vs. control). BemA samples showed an inhibition of the classical pathway for bile acid synthesis that is controlled by Cyp7a1 activity, while there were no modifications in the alternative pathway controlled by Cyp27a1 (see Figure 3 and Figure 4). Thus, the BemA-related increase in the expression of *cyp8b1* could facilitate the synthesis of bile acids and their delivery to the ileum lumen, increasing the activation of ileal FXR and the production of FGF15/19. Rodent bile acids are mainly excreted into the digestive tract in the form of tauro-conjugates [19]. In this sense, it is interesting to note that the liver expression of the taurine transporter *slc6a6* was also decreased in the HFHFr liver samples (×0.15), while BemA treatment restored *slc6a6* expression to control values (see Figure 4).

### 2.3. BemA Administration Improves the Ratio of Phophor-Ser^571^/Total PGC1α (Peroxisome Proliferator Activator Receptor Gamma Coactivator-1α) in the Livers of HFHFr-Fed Female Sprague-Dawley Rats

PGC1α acts as a transcriptional coactivator for a wide array of transcription factors and nuclear receptors, including FXR [20]. We looked at the hepatic protein (Figure 6A) and mRNA (Figure 6D) levels of PGC1α in our liver samples, as well as the levels of phophor-Ser^571^ PGC1α (Figure 6B), an inactive form of PGC1α. Although there were no significant changes in the relative levels of *pgc-1α* mRNA, both experimental interventions (HFHFr and BemA) increased the total PGC1α (×2.0 and ×5.9, respectively) and phophor-Ser^571^ PGC1α (×2.0 and ×4.1, respectively) protein levels. The ratio of phosphorylated to total PGC1α protein level (Figure 6C) was reduced by BemA treatment (×0.5 vs HFHFr), partially reversing the blockade of PGC1α coactivator activity in the HFHFr liver samples. Thus, while practically all the PGC1α protein in the HFHFr samples was in the phosphorylated, inactive form, BemA treatment restored the ratio of inactive/phosphorylated to total PGC1α protein level to control values, allowing PGC1α to coactivate FXR transcriptional activity.

Finally, we determined the liver expression of several kinases that can reportedly phosphorylate PGC1α at the Ser^571^ position, such as Akt serine/threonine kinase, CDC-like kinase 2, and S6K1 (ribosomal protein S6 kinase beta-1) [20]. Of these, only S6K1 activity, indirectly ascertained through its degree of phosphorylation at the Thr^389^ position (see Figure 7), was in accordance with the pattern of phophor-Ser^571^ PGC1α expression, with its phosphorylated, active form being significantly reduced (×0.5 vs. HFHFr) by BemA administration.

S6K1 is downstream of mTORC1 signaling [21]. Thus, we indirectly determined, in the same liver samples, the degree of mTORC1 activation by comparing the relative amounts of phophor-Ser^2481^ mTORC1. As can be seen in Figure 8, the HFHFr diet did not modify the amount of phophor-Ser^2481^ mTORC1 in the liver samples with respect to control values. BemA treatment significantly reduced mTORC1 phosphorylation compared to the control (×0.24) and HFHFr liver samples (×0.39), which is consistent with the abovementioned reduction in the levels of phosphor-Thr^389^ S6K1.

## 3. Discussion

In the presented work, we show that the administration of BemA to female Sprague-Dawley rats fed an HFD supplemented with a 10% *w*/*v* fructose solution as a beverage (HFHFr diet) reverses the decreased production of liver H_2_S induced by consumption of the HFHFr diet. As H_2_S has a prominent role in the regulation of liver glucose and triglyceride metabolism [22], the recovery of liver H_2_S production induced by BemA could contribute to the recently described therapeutic effect of BemA in several experimental models of NAFLD [18,23].

It has been previously shown that the ingestion of liquid fructose reduces the expression of hepatic CSE, which controls liver H_2_S synthesis [12]. Furthermore, rodent diets rich in saturated fatty acids also reduce the liver expression of CSE in mice [6]. Thus, as BemA administration significantly reduced total calorie intake in our HFHFr experimental model (×0.88 vs. HFHFr, see [18]), it could be argued that the effect of BemA on liver CSE expression and H_2_S production is merely a consequence of the reduced ingestion of the HFHFr diet, as reflected by the reduced calorie intake shown by BemA-treated rats. However, the BemA-treated rats ingested the same amount of solid HFD chow as the HFHFr-fed rats (8.1 ± 1.0 and 7.9 ± 0.8 g/rat/day, respectively). Although they significantly reduced their ingestion of liquid fructose with respect to the HFHFr group, they still consumed a considerable amount of fructose during the BemA treatment period (almost 500 fructose-derived kcal per rat). Thus, these data do not justify a recovery in liver CSE expression and H_2_S production based only on changes in the amount of saturated fatty acids and/or fructose consumed, implying a specific effect of BemA on the regulation of H_2_S metabolism. Consequently, we searched for possible molecular mechanisms to explain the BemA-induced changes in liver CSE expression.

Our previous work already provided two possible and complementary mechanisms. First, an increase in the liver expression and activity of the transcription factor SREBP1c has been indirectly linked to a decrease in CSE expression, through miR216a production [6]. The reduction in SREBP1c expression and activity observed in the liver samples of BemA-treated rats [18] could indirectly result in an increase in CSE expression and, consequently, in liver H_2_S production. Second, the activity of the PERK-ATF4 (activating transcription factor 4) branch of the ERSR is directly involved in regulating the expression of hepatic CSE [24]. We have previously shown that the HFHFr diet reduces the liver levels of the phosphor-Thr^981^ PERK protein, a marker of PERK activity [16,17]. When data on the expression of liver phosphor-Thr^981^ and total PERK protein shown in Bentanachs et al. [17] is presented as the ratio of phosphor/total PERK protein, the liver samples from BemA-treated rats showed a significant increase in the ratio when compared to the HFHFr group (1.00 ± 0.19, 0.77 ± 0.15, and 1.36 ± 0.26 for control, HFHFr, and BemA, respectively; *p* < 0.01 BemA vs. HFHFr, ×2.14), implying that BemA treatment could restore liver PERK activity and, therefore, CSE expression.

Data in the present work indicate that BemA treatment can also increase liver CSE expression and H_2_S production by restoring liver FXR transcriptional activity by reducing the amount of the inactive, phosphorylated form of its coactivator, PGC1α. It is well known that the transcription of the liver *pgc1α* gene is increased during prolonged fasting [25]. Unexpectedly, both the HFHFr diet and BemA administration significantly increased the amount of PGC1α protein in the liver samples. Although we do not know at this moment the molecular mechanism responsible for this effect, it is not related, as in the case of fasting, to increased transcription of the liver *pgc1α* gene, as the relative amount of PGC-1α mRNA was unchanged in both experimental conditions. Nevertheless, practically all the PGC1α protein present in the liver samples from the HFHFr group was present as phophor-Ser^571^ PGC1α, an inactive form of PGC1α [20,26]. Consequently, FXR transcriptional activity was impaired in the livers of the HFHFr-fed rats. On the contrary, in the livers of the BemA-treated rats, the relative amount of phophor-Ser^571^ PGC1α was reduced with respect to HFHFr values, allowing sufficient PGC1α coactivator activity, and thus, an increase in FXR transcriptional activity. Furthermore, our data show that BemA restored the liver activity of the PGC1α/FXR pathway by inhibiting the activity of liver mTORC1, thus reducing the enzymatic activity of the kinase S6K1, a downstream effector of mTORC1 signaling [21] that is responsible for the phosphorylation and inactivation of PGC1α (Figure 9).

Our work unravels a possible new therapeutic mechanism of BemA: the inhibition of mTORC1 activity. The inhibition of mTORC1 and its downstream effector S6K1 mimics the metabolic response to caloric restriction, with both situations reducing blood cholesterol, body weight, and fat mass in humans and in rodents [27]. In fact, in our previous work using the same experimental procedure [18], we found that BemA administration to HFHFr-fed rats reduced body weight and the weight of the perigonadal and subcutaneous WAT (white adipose tissue) without changing the expression of thermogenic markers in WAT, and even decreasing them in brown adipose tissue. Thus, the BemA-related reduction in mTORC1-S6K1 signaling could also be responsible for the BemA-related effects on WAT and body weight. However, as we have previously reported that liquid fructose increases mTORC1 activity in female Sprague-Dawley rats [28] and given that BemA inhibits fructokinase expression and liquid fructose ingestion [18], we cannot discard an indirect effect of BemA on mTORC1 activity through a reduction in liquid fructose consumption.

## 4. Material and Methods

### 4.1. Animals and Experimental Design

Two-month-old female Sprague-Dawley rats weighing 178 ± 8 g (Envigo, Barcelona, Spain) were housed two per cage under conditions of constant humidity (40–60%) and temperature (20–24 °C), with a light/dark cycle of 12 h. Twenty-four rats were randomly assigned into three groups (n = 8 each) and fed the diets for 3 months: (i) the control group (CT) was fed a regular chow diet (2018 Teklad Global rodent diet) with free access to water; (ii) the high-fat high-fructose group (HFHFr) was fed a high-fat diet and had free access to a 10% *w*/*v* fructose solution; and (iii) the bempedoic acid group (BemA) was fed the HFHFr diet and was treated orally with 30 mg/kg/day of BemA (MedChemTronica, Sollentuna, Sweden) by gavage during the third month of treatment. For more details, please see our previous publication [18].

### 4.2. Sample Preparation

At the end of the treatment, rats fasted for 2 h and were anesthetized with ketamine/xylazine (9 mg/40 µg per 100 g of body weight, respectively), and blood was collected into microtubes (Sarstedt AG & Co., Nümbrecht, Germany) through cardiac puncture and centrifuged at 10,000× *g* for 5 min at room temperature. Rats were euthanized by exsanguination, and the livers were perfused, immediately frozen in liquid nitrogen, and stored at −80 °C until needed for biomolecular assays.

### 4.3. Serum Analytes

Total serum bile acids were determined using the specific colorimetric kit from SpinReact (Girona, Spain). FGF15/19 serum levels were analyzed by MyBioSourse (San Diego, CA, USA) ELISA kit, MBS 2025580.

### 4.4. Liver Bile Acids Determination

For each sample, 100 mg of liver tissue in 1.2 mL phosphate-buffered saline (PBS) were homogenized and centrifuged (10,000× *g*, 30 min), and the supernatants were used to measure total liver bile acids using the specific colorimetric kit from SpinReact (Girona, Spain).

### 4.5. Determination of H_2_S Production by Liver Tissue

H_2_S production in liver tissue was evaluated following the lead sulfide method as previously described, with modifications [29]. Briefly, 100 mg of liver tissue were homogenized in 1.2 mL PBS and their protein levels measured with the BCA Assay Kit (Thermo Fisher, Waltham, MA, USA). For total H_2_S production measurement, 200 μg of protein were incubated at 37 °C in the presence of 10 mM Cys (Sigma-Aldrich, St. Louis, MO, USA) and 20 μM Pyridoxal 5′-phosphate (PLP) (Sigma-Aldrich, St. Louis, MO, USA) on 96-well plates covered with a lead acetate membrane. In the case of enzymatic H_2_S production, 15 mM EDTA solution was added to the previous reaction mix to chelate Fe^2+^, a metal used in the non-enzymatic production, according to Yang et al. [30]. Incubations were performed after 2 h, until dots of lead sulfide were detected but not saturated. In order to prepare the membranes, Whatman n° 2 paper was soaked in 20 mM lead acetate (Sigma-Aldrich, St. Louis, MO, USA) and vacuum dried. Dots were densitometered (BioRad Densitometer G-800, Hercules, CA, USA) for quantification. A standard curve from 0 to 1000 mM NaHS was performed for each membrane.

### 4.6. RNA Extraction and Quantitative RT-qPCR Analysis

Total RNA was isolated from livers (50~100 mg of each sample) by using the Trisure^®^ reagent (Bioline, Meridian Biosciences, Cincinnati, OH, USA) according to the manufacturer’s instructions. RNA concentration was determined by measuring absorbance at 260 nm, while the 260/280 nm absorbance ratio was used to analyse RNA quality.

For the real-time polymerase chain reaction (RT-qPCR), 1 μg RNA was reverse transcribed into cDNA using the Moloney Murine Leukemia Virus Reverse Transcriptase (MLV-RT; Invitrogen, Carlsbad, CA, USA), and the specific mRNAs were assessed in the StepOnePlus Real-Time PCR System Thermal Cycling Block (Applied Biosystems, Foster City, CA, USA), using 100 µM of each specific primer, 10 ng of cDNA, and SYBR^®^ Green PCR Master Mix (Applied Biosystems, Thermo Fisher Scientific, Waltham, MA, USA) in 96-well plates. mRNA expression was calculated using the recommended 2^−ΔΔCt^ method. The *β-actin* and RPS29 were used as the housekeeping genes to normalize the results. The primer sequences, Genbank TM number, and PCR product lengths are listed in the Appendix A.

### 4.7. Protein Extraction and Western Blot

Liver samples were homogenized in lysis buffer using a Potter-Elvehjem homogenizer, and total protein and nuclear extracts were prepared as described previously [16]. Protein content was determined by the Bradford assay [31]. Western blots were performed using four samples per group, each sample pooled from two animals. A total of 20–30 µg of protein extracts was subjected to SDS-polyacrylamide gel electrophoresis. Proteins were then transferred onto Immobilon polyvinylidene difluoride transfer membranes (Millipore, Billerica, MA, USA) and blocked for 1 h at room temperature with a 5% non-fat milk solution in Tris-buffered saline (TBS) containing 0.1% Tween-20. Membranes were then incubated with specific primary antibodies (see the list of antibodies used in the Appendix A). Detection was performed using the Immobilion Western HRP substrate Peroxide Solution^®^ (Millipore, Billerica, MA, USA). To confirm the uniformity of protein loading, blots were incubated with antivinculin antibody (Santa Cruz Biotech, Dallas, TX, USA) as a control for total protein extracts, and with anti-TBP antibody (AbCam, Cambridge, UK) as a control for nuclear protein extracts.

### 4.8. Statistical Analysis

Results are expressed as mean ± standard deviation (SD). Significant differences were established by one-way ANOVA and Tukey’s post-hoc test (GraphPad Software version 9, San Diego, CA, USA). When the SD of the group was different according to the Brown-Forsythe test, the data were transformed into their logarithms and ANOVA was rerun. The level of statistical significance was set at *p* < 0.05.

## 5. Conclusions

We demonstrate that BemA administration to HFHFr-fed rats restores liver H_2_S production, as well as CSE and CBS expression, through several mechanisms, including the reduction of liver expression and activity of SREBP1c, the induction of liver expression and activity of PERK, and an increase in liver FXR activity, probably through the inhibition of the mTORC1-S6K1 pathway.

## Figures and Tables

**Figure 1 ijms-24-00473-f001:**
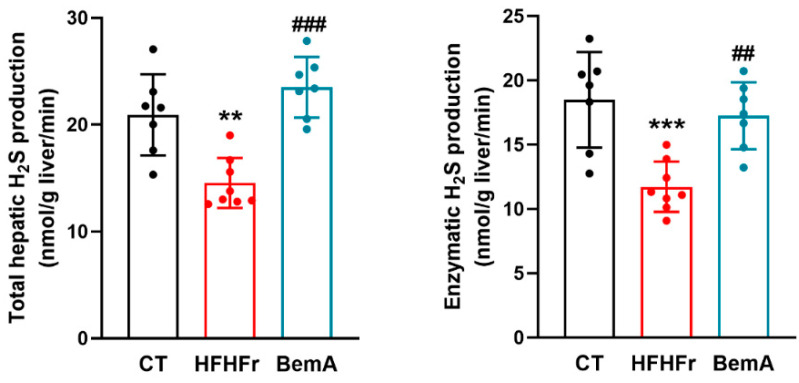
Effect of HFHFr supplementation and BemA treatment on Sprague-Dawley female rats’ liver total and enzymatic H_2_S production. Bar plots show liver total and enzymatic H_2_S production as mean ± SD, as well as individual values from 7–8 tissue samples, corresponding to the three experimental groups studied: CT (control), HFHFr (high-fat high-fructose), and BemA (HFHFr treated with BemA) female Sprague-Dawley rats. ** *p* < 0.01, *** *p* < 0.001 vs. CT; ^##^
*p* < 0.01, ^###^
*p* < 0.001 vs. HFHFr.

**Figure 2 ijms-24-00473-f002:**
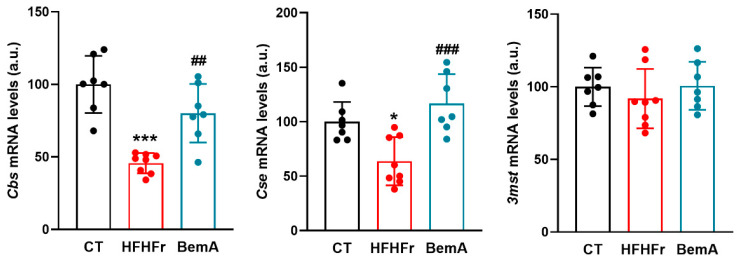
Relative expression of the three enzymes involved in the release of H_2_S in liver tissue: *cbs*, *cse*, and *3mst*. Bar plots show the relative mRNA levels of *cbs*, *cse*, and *3mst* genes as mean ± SD, as well as individual values from 7-8 tissue samples, corresponding to the three experimental groups studied: CT (control), HFHFr (high-fat high-fructose), and BemA (HFHFr treated with BemA) female Sprague-Dawley rats. * *p* < 0.05, *** *p* < 0.001, vs. CT; ^##^
*p* < 0.01, ^###^
*p* < 0.001 vs. HFHFr.

**Figure 3 ijms-24-00473-f003:**
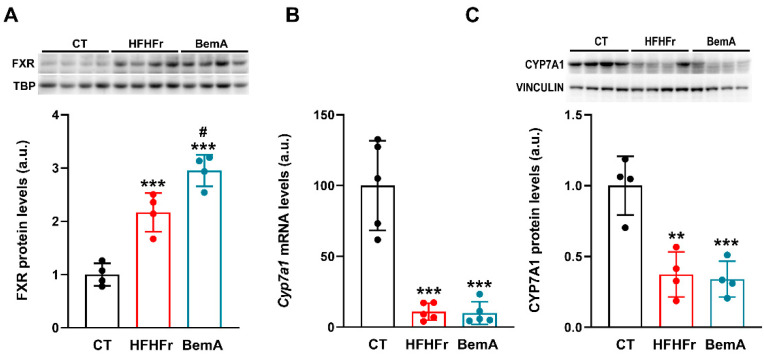
Effect of Bempedoic acid on nuclear FXR protein levels and Cyp7A1 expression in liver samples. Bar plots show (**A**) liver FXR protein levels in nuclear extracts from control (CT), high-fat high-fructose (HFHFr), and bempedoic acid (BemA)-treated rats. The results are expressed as the mean ± SD of four pooled samples, obtained by mixing equal amounts of tissue from two rats, for each condition. (**B**) Bar plots show mRNA and (**C**) protein levels of Cyp7A1 in liver samples from CT, HFHFr, and BemA rats. The results are expressed as the mean ± SD of five samples (mRNA) or four pooled samples (WB), obtained by mixing equal amounts of tissue from two rats, for each condition. On the upper side of the figure, representative western blot bands corresponding to the three different study groups are shown. a.u.: arbitrary units. Data were analyzed by one-way ANOVA followed by Tukey’s post-hoc test. ** *p* < 0.01, *** *p* < 0.001 vs. CT; ^#^
*p* < 0.05 vs. HFHFr.

**Figure 4 ijms-24-00473-f004:**
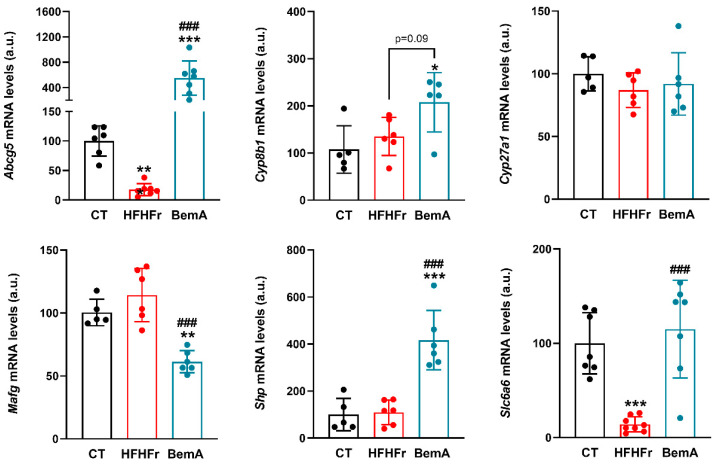
Relative expression of several genes under FXR transcriptional control in liver samples. Bar plots show the relative mRNA levels of *abcg5*, *cyp8b1*, *cyp27a1*, *mafg*, *shp*, and *slc6a6* genes as mean ± SD, as well as individual values from 5-8 tissue samples, corresponding to the three experimental groups studied: CT (control), HFHFr (high-fat high-fructose), and BemA (HFHFr treated with BemA) female Sprague-Dawley rats. * *p* < 0.05, ** *p* < 0.01, *** *p* < 0.001, vs. CT; ^###^
*p* < 0.001 vs. HFHFr.

**Figure 5 ijms-24-00473-f005:**
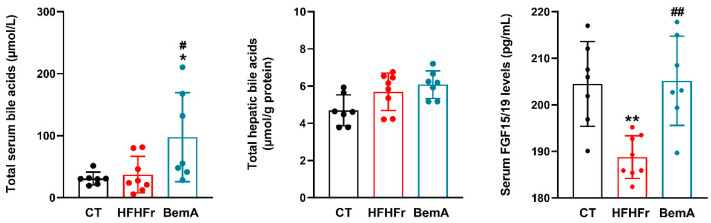
Effect of HFHFr supplementation and BemA treatment on Sprague-Dawley female rats’ serum and liver bile acid concentrations and serum FGF15/19 levels. Bar plots show concentrations as mean ± SD, as well as individual values from 7–8 samples, corresponding to the three experimental groups studied: CT (control), HFHFr (high-fat high-fructose), and BemA (HFHFr treated with BemA) female Sprague-Dawley rats. * *p* < 0.05, ** *p* < 0.01 vs. CT; ^#^
*p* < 0.05, ^##^
*p* < 0.01 vs. HFHFr.

**Figure 6 ijms-24-00473-f006:**
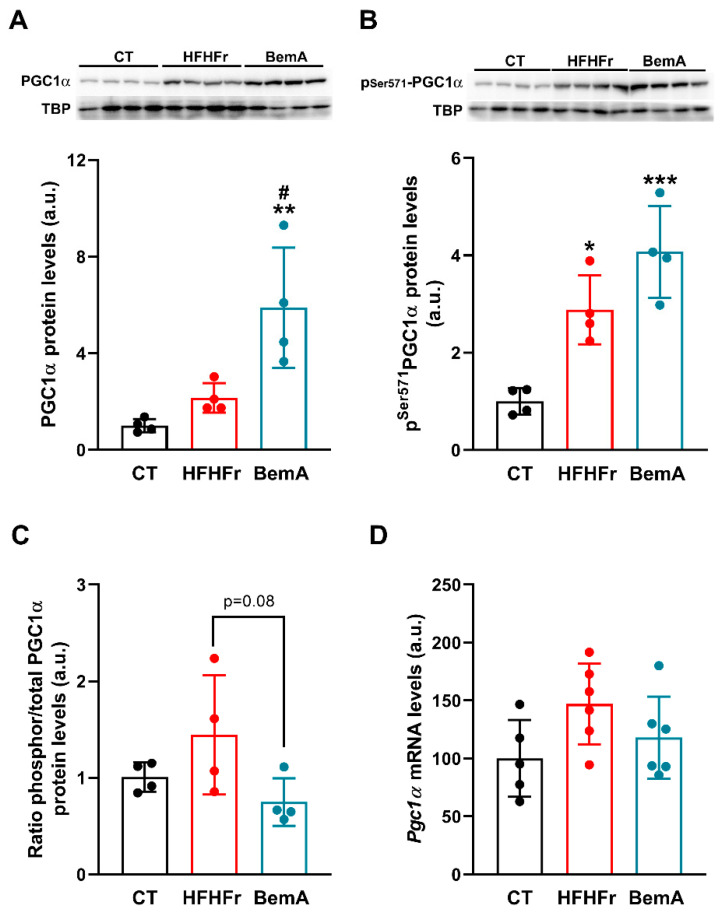
Effect of HFHFr supplementation and BemA treatment on Sprague-Dawley female rats’ liver expression of PGC1α. Bar plots show total (**A**) and phosphor-Ser^571^ (**B**) liver protein levels from control (CT), high-fat high-fructose (HFHFr), and bempedoic acid (BemA)-treated rats. The results are expressed as the mean ± SD of four pooled samples, obtained by mixing equal amounts of tissue from two rats, for each condition. On the upper side of the figures, representative western blot bands corresponding to the three different study groups are shown; (**C**) bar plots show the ratio of phosphor-Ser^571^ to total PGC1α protein levels for each pooled sample shown in Figure 6A,B; (**D**) bar plots show the relative mRNA levels of the *pgc1α* gene as mean ± SD, as well as individual values from 5–6 tissue samples, corresponding to the three experimental groups studied. a.u.: arbitrary units. Data were analyzed by one-way ANOVA followed by Tukey’s post-hoc test. * *p* < 0.05, ** *p* < 0.01, *** *p* < 0.001 vs. CT; # *p* < 0.05 vs. HFHFr.

**Figure 7 ijms-24-00473-f007:**
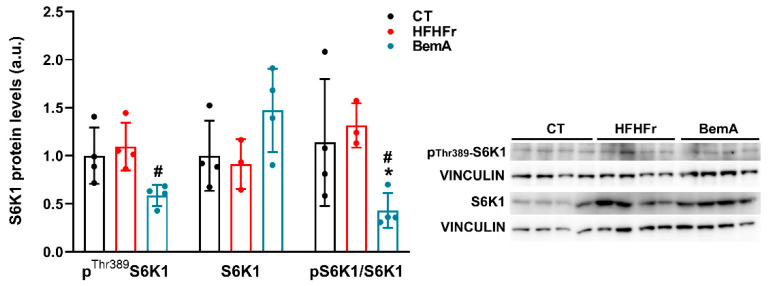
Effect of Bempedoic acid on S6K1 protein levels in liver samples. Bar plots show liver phosphor-Thr^389^ S6K1, total S6K1 protein levels, and the ratio of phosphor to total S6K1 protein levels from control (CT), high-fat high-fructose (HFHFr), and bempedoic acid (BemA)-treated rats. The results are expressed as the mean ± SD of four pooled samples, obtained by mixing equal amounts of tissue from two rats, for each condition. On the right side of the figure, representative western blot bands corresponding to the three different study groups are shown. a.u.: arbitrary units. Data were analyzed by one-way ANOVA followed by Tukey’s post-hoc test. * *p* < 0.05 vs. CT; ^#^
*p* < 0.05 vs. HFHFr.

**Figure 8 ijms-24-00473-f008:**
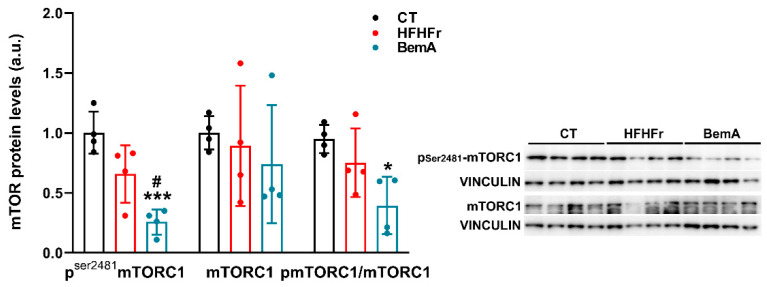
Effect of Bempedoic acid on mTORC1 protein levels in liver samples. Bar plots show liver phosphor-Ser^2481^ mTORC1, total mTORC1 protein levels, and the ratio of phosphor to total mTORC1 protein levels from control (CT), high-fat high-fructose (HFHFr), and bempedoic acid (BemA)-treated rats. The results are expressed as the mean ± SD of four pooled samples, obtained by mixing equal amounts of tissue from two rats, for each condition. On the right side of the figure, representative western blot bands corresponding to the three different study groups are shown. a.u.: arbitrary units. Data were analyzed by one-way ANOVA followed by Tukey’s post-hoc test. * *p* < 0.05, *** *p* < 0.001 vs. CT; ^#^
*p* < 0.05 vs. HFHFr.

**Figure 9 ijms-24-00473-f009:**
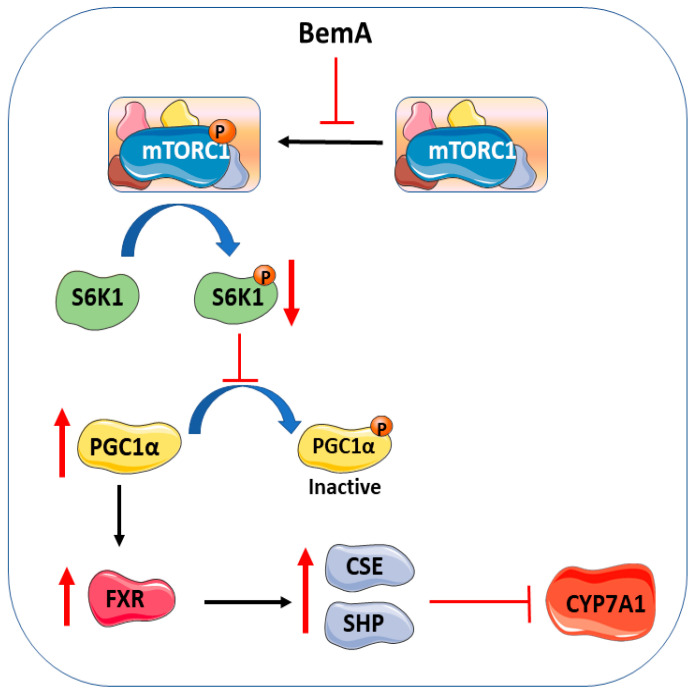
BemA’s effect on the mTORC1/S6K1/PGC1α pathway.

## Data Availability

The data presented in this study is contained within the article. Raw data is available as Appendix A.

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
