# Peer review of "Bempedoic Acid Restores Liver H2S Production in a Female Sprague-Dawley Rat Dietary Model of Non-Alcoholic Fatty Liver"

_ijms, 2022, doi:10.3390/ijms24010473_

Round 1

Reviewer 1 Report

In this manuscript the authors study the effect of bempedoic acid (BemA) on the liver metabolism of rats that had been fed a high-fat-high-fructose (HFHFr) diet. They show a number of difficult metabolic effects at the molecule level and present a model to explain the restoration of H2S production in the livers of these animals. Some of the effects noted were:

1) restored total and enzymatic H2S production

2) restored expression of cbs and cse

3) Increased FXR levels and decreased CYP7A1 levels

4) reduced phospho-mTORC1 levels

This paper is an excellent piece of work. The authors have put a lot of work into evaluating the effects of BemA on liver metabolism. I especially appreciated figure 9 when they attempt to construct the model of BemA effect on the pathway.

At the end of the paper the authors state that "we cannot discard an indirect effect of BemA on mTORC1 activity through a reduction of liquid fructose consumption". I assume this statement means that the rats were consuming less liquid fructose. If this is the case, is this not easy to measure by observing the amount of liquid the rats consume?

Author Response

We appreciate very much the positive comments offered by reviewer 1 about our work. As the reviewer points out, we indeed have measured the amount of liquid fructose consumed by rats. From the original data of liquid fructose ingestion (reported in our previous publication Velázquez et al., Biomedicines 2022, 10, 1517, see Figure 2), we have calculated that the administration of BemA reduced the intake of fructose-derived calories by 50% on average with respect to HFHFr rats. Nevertheless, BemA-treated rats still consumed an appreciable amount of fructose during the treatment period (around 500 fructose-derived kcal per rat). This information is provided in the discussion section, lines 312-313, of the present manuscript.  

Reviewer 2 Report

Authors demonstrated the successful restoration of H2S in fatty liver mice model by treating with bempedoic acid. The work is novel and could be useful for further research in fatty liver disease. BemA treatment restored H2S production, increasing the expression of CBS and CSE, providing evidence for the involvement of FXR transcriptional activity, and the mTORC1 /S6K1 / PGC1α pathway which looks promising finding.

Authors are suggested to do the following modifications:

 1.      Would it be more suitable to mention fatty liver as non-alcoholic fatty liver in the article title? It will be easy to understand which type of fatty liver you are talking about.

2.      Typos: Sentence 94, period (authors wrote preriod)

3.      Did you perform any gender comparison study initially, why choosing female mice only?

4.      In figure 7, can you replace western blot figure with better one? S6K1, VINCULIN bands are kind of curved, better-quality image would be perfect.

5.      In figure 8, mTORC1 bands are quite broad and hazy. Do you expect the phosphorylated mTORC1?

Author Response

We appreciate very much the appraisal of our work forwarded by reviewer 2 and the constructive suggestions provided.

Sugg. 1:  Would it be more suitable to mention fatty liver as non-alcoholic fatty liver in the article title? It will be easy to understand which type of fatty liver you are talking about. We agree that the title would provide a more precise information if we include “non-alcoholic” on it, and consequently, we have modified the title in the revised version of the manuscript.

Sugg. 2: Sentence 94, period (authors wrote preriod). The typographic error has been corrected.

Sugg.3.:  Did you perform any gender comparison study initially, why choosing female mice only? In the last two decades, we have been involved in the study of the metabolic effects of fructose in rodents. In 2011, we reported that female Sprague-Dawley rats were more responsive to the deleterious effects of fructose than males (Vilà et al., J Nutr Biochem 22 (2011) 741–751). For this reason, we consistently use female Sprague-Dawley rats in our experimental models. At this moment, we are conducting experiments to ascertain possible metabolic differences, if any, between male and female rats supplemented with the HFHFr diet.

Sugg.4: In figure 7, can you replace western blot figure with better one? S6K1, VINCULIN bands are kind of curved, better-quality image would be perfect. We agree that in Figure 7, the fourth control sample shows tilted bands for total S6K1 and vinculin, but their shape is sharp and they can be correctly quantified, given similar values than the other control samples. We are sorry, but we do not have enough samples left and the best WB we have obtained is the one shown in Figure 7.

Sugg.5: In figure 8, mTORC1 bands are quite broad and hazy. Do you expect the phosphorylated mTORC1? We also agree that these bands are broad and hazy, but they are the best we could obtain; in any case, as expected, they do not show marked differences among the three experimental groups studied. Regarding the phosphorylation of mTORC1, as we detected previously that the phosphorylation of S6K1 was unmodified in the HFHFr group, we did not expect a significant increase in phosphorylated mTORC1 in our HFHFr samples.